# Safety of Multiple Vaccinations and Durability of Vaccine-Induced Antibodies in an Italian Military Cohort 5 Years after Immunization

**DOI:** 10.3390/biomedicines10010006

**Published:** 2021-12-21

**Authors:** Claudia Ferlito, Vincenzo Visco, Roberto Biselli, Maria Sofia Cattaruzza, Giulia Carreras, Gerardo Salerno, Florigio Lista, Maria Rosaria Capobianchi, Concetta Castilletti, Daniele Lapa, Guido Antonelli, Massimo Gentile, Maurizio Sorice, Gloria Riitano, Giuseppe Lucania, Valeria Riccieri, Fabrizio Mainiero, Antonio Angeloni, Marco Lucarelli, Giampiero Ferraguti, Alberto Autore, Marco Lastilla, Simonetta Salemi, Michela Ileen Biondo, Andrea Picchianti-Diamanti, Sara Caporuscio, Raffaela Teloni, Sabrina Mariotti, Roberto Nisini, Raffaele D’Amelio

**Affiliations:** 1Dipartimento di Medicina Clinica e Molecolare, Sapienza Università di Roma, Via di Grottarossa 1035-1039, 00189 Roma, Italy; clau.ferlito@gmail.com (C.F.); vincenzo.visco1@uniroma1.it (V.V.); gerardo.salerno@uniroma1.it (G.S.); simonetta.salemi@uniroma1.it (S.S.); biondo_michela@yahoo.it (M.I.B.); andrea.picchiantidiamanti@uniroma1.it (A.P.-D.); sara.caporuscio1@gmail.com (S.C.); raffaele.damelio@gmail.com (R.D.); 2Ispettorato Generale della Sanità Militare, Stato Maggiore della Difesa, Via S. Stefano Rotondo 4, 00184 Roma, Italy; bobbiselli@libero.it; 3Dipartimento di Sanità Pubblica e Malattie Infettive, Sapienza Università di Roma, Piazzale Aldo Moro 5, 00185 Roma, Italy; mariasofia.cattaruzza@uniroma1.it; 4Istituto per lo Studio la Prevenzione e la Rete Oncologica, Via Cosimo il Vecchio 2, 50139 Firenze, Italy; g.carreras@ispro.toscana.it; 5Dipartimento Scientifico Policlinico Militare di Roma, Esercito Italiano, Via S. Stefano Rotondo 4, 00184 Roma, Italy; florigio.lista@esercito.difesa.it; 6Laboratorio di Virologia, IRCCS, Istituto Nazionale Malattie Infettive “Lazzaro Spallanzani”, Via Portuense 292, 00149 Roma, Italy; maria.capobianchi@inmi.it (M.R.C.); concetta.castilletti@inmi.it (C.C.); daniele.lapa@inmi.it (D.L.); 7Dipartimento di Medicina Molecolare, Sapienza Università di Roma, Viale di Porta Tiburtina 28, 00185 Roma, Italy; guido.antonelli@uniroma1.it (G.A.); massimo.gentile@uniroma1.it (M.G.); 8Dipartimento di Medicina Sperimentale, Sapienza Università di Roma, AOU Policlinico Umberto I, Viale del Policlinico 155, 00161 Roma, Italy; maurizio.sorice@uniroma1.it (M.S.); gloria.riitano@uniroma1.it (G.R.); giuseppe.lucania@uniroma1.it (G.L.); fabrizio.mainiero@uniroma1.it (F.M.); antonio.angeloni@uniroma1.it (A.A.); marco.lucarellii@uniroma1.it (M.L.); giampiero.ferraguti@uniroma1.it (G.F.); 9Dipartimento di Scienze Cliniche Internistiche, Anestesiologiche e Cardiovascolari, Sapienza Università di Roma, AOU Policlinico Umberto I, Viale del Policlinico 155, 00161 Roma, Italy; valeria.riccieri@uniroma1.it; 10Osservatorio Epidemiologico della Difesa, Ispettorato Generale della Sanità Militare, Stato Maggiore della Difesa, Via S. Stefano Rotondo 4, 00184 Roma, Italy; alberto.autore@aeronautica.difesa.it (A.A.); marco.lastilla@aeronautica.difesa.it (M.L.); 11Dipartimento di Malattie Infettive, Istituto Superiore di Sanità, Viale Regina Elena 299, 00161 Roma, Italy; raffaela.teloni@iss.it (R.T.); sabrina.mariotti@iss.it (S.M.)

**Keywords:** vaccines, safety, antibody durability, antibody persistence, B-cell polyclonal activation

## Abstract

We previously examined the safety and immunogenicity of multiple vaccines administered to a military cohort, divided into two groups, the first composed of students at military schools, thus operating inside the national borders for at least 3 years, and the other formed of soldiers periodically engaged in a 9-month-long mission abroad (Lebanon). In the current study, we analyzed 112 individuals of this cohort, 50 pertaining to the first group and 62 to the second group, in order to examine the possible late appearance of side effects and to calculate the half-life of the induced antibodies. Moreover, the possible involvement of B-cell polyclonal activation as a pathogenetic mechanism for long term antibody persistence has even been explored. No late side effects, as far as autoimmunity and/or lymphoproliferation appearance, have been noticed. The long duration of the vaccine induced anti-HAV antibodies has been confirmed, whereas the antibodies induced by tetravalent meningococcal polysaccharide vaccine have been found to persist above the threshold for putative protection for a longer time, and anti-tetanus, diphtheria, and polio 1 and 3 for a shorter time than previously estimated. No signs of polyclonal B-cell activation have been found, as a possible mechanism to understand the long antibody persistence.

## 1. Introduction

Military personnel are especially exposed to the risk of infectious diseases, due to their daily activity, which exposes them to trauma and contaminated wounds, and community life often in extremely unfavorable environmental conditions, thus they are required to have mandatory vaccinations all over the world [1]. The military, therefore, is the adult population with the highest experience of vaccinations. However, despite the undeniable merits that the military has in the prevention and control of infectious diseases, frequently even in favor of the general population, relatively few studies have been carried out on the possible reciprocal interference by contemporaneously administered multiple different vaccines regarding vaccine immunogenicity and effectiveness as well as on the possible induction of side effects. Such a relative scarcity of systematic studies has allowed that poorly understood phenomena, such as the Gulf War Syndrome in the USA [2], and the occurrence of lymphoproliferative diseases in the young military in Italy could be ascribed to a consequence of simultaneously administered multiple vaccinations [3].

In the attempt to provide a further contribution in this field, we analyzed a military cohort composed of two different groups, group 1 was formed of military students at the military schools, thus newly enrolled and vaccinated, and stably operating in the national borders for at least 3 years, corresponding to the minimum length of the training period, and group 2 of older military individuals, periodically engaged in a 9-month operational activity abroad (Lebanon). Previous studies have allowed establishing that no autoimmune/lymphoproliferative phenomena could be observed up to 9-month post-vaccination in both groups [4]; moreover, the specific immune response to meningococcal tetravalent polysaccharide vaccine [5], tetanus/diphtheria [6] and viral vaccines, including hepatitis A virus (HAV), measles, mumps, and rubella (MMR), polio and influenza [7] was excellent and long durable.

The aim of the current study was the additional control of immune parameters after a longer follow-up to 5-year in some individuals of the cohort already studied and followed-up to 9-month after a multiple vaccination schedule.

The possible appearance of late side effects has been investigated, as well as the duration of the antibody response at the putative protective level has been calculated and checked. Finally, the possible contribution of the B-cell polyclonal activation to the antibody level persistence has also been tested.

## 2. Materials and Methods

### 2.1. Study Population

As previously reported [4], from September 2012 to June 2014 two groups of Italian military personnel, the first (group 1) represented by newly recruited students at military schools residing in Italy for at least 3 years and the other (group 2) by soldiers operating abroad (Lebanon) for nine months, were enrolled. Exclusion criteria were pregnancy, immunodepression, vaccine hypersensitivity. For the parameters evaluated in the current study blood samples were collected in both groups in 2017. This time was denominated T3, and it was compared with the same parameters collected in the same subjects at T2 (nine months following the vaccine administration).

The study was approved by the Italian Ministry of Defense ethical committee in July 2011 and registered in clinicaltrials.gov in 2012 with the identifier NCT01807780.

### 2.2. Vaccination Schedule

As previously reported [4], at enrollment, informed consent and the medical history of all individuals were collected. The vaccination schedule was personalized based on the history of infectious diseases and already received vaccinations; moreover, the vaccination schedule was even tailored on type of employment, either national or abroad. Typhoid vaccine was only administered to the military of group 2. Administered vaccines were the following: tetanus/diphtheria (Td, DifTetAll-Novartis Vaccines and Diagnostics, Siena, Italy), inactivated polio (Imovax polio-Sanofi Pasteur MSD SpA, Roma, Italy), measles/mumps/rubella (MMR, Priorix-GlaxoSmithKline SpA, Verona, Italy), chickenpox (Varivax-Sanofi Pasteur MSD SpA, Roma, Italy) polysaccharide tetravalent (A, C, W_135_, Y) meningococcal meningitis (Mencevax-Pzifer Srl, Latina, Italy), hepatitis A virus (HAV, Epaxal-Crucell Italy Srl, Baranzate, Italy), hepatitis B virus (HBV, Engerix B-GlaxoSmithKline SpA, Verona, Italy), influenza (Fluad-Seqirus Srl, Siena, Italy), and typhoid (Vivotif Berna-PaxVax Ltd., Birmingham, UK). Vaccines were generally administered the same day (in different arms), but in a few cases up to two weeks apart.

### 2.3. Safety, Immunogenicity, and Effectiveness

Vaccine safety was examined by the military monitoring system of vaccination adverse events and by the evaluation of the peripheral blood cell count, serum protein electrophoresis and serum immunoglobulins (Ig), to monitor the possible onset of signs suggestive of lymphoproliferative disorders, as previously reported [4]. Moreover, the search for autoantibodies (anti-nuclear [ANA], anti-double-stranded DNA [anti-dsDNA], anti-extractable nuclear antigens [anti-ENAs], anti-phospholipid [APL], anti-neutrophil cytoplasmic antibodies [ANCA] and rheumatoid factor [RF]), was performed. Immunogenicity was carried out by examining the levels of anti-vaccine antibodies. This further point was used in the calculation of the antibody duration above the putative threshold for protection. Lastly, effectiveness was determined through the military surveillance system by monitoring the number of cases of infectious diseases against which the vaccines should induce protection.

### 2.4. Autoantibodies

As previously reported [4], ANA, anti-dsDNA and ANCA were detected by indirect immunofluorescence (IIF), using Hep-2 cells, *Crithidia luciliae* slides, formalin- and ethanol-fixed neutrophils as substrate, respectively. Titers ≥ 1:80 (for ANA), ≥1:20 (for anti-dsDNA) and ≥1:20 (for ANCA) were considered positive. Anti-ENAs were identified by ELISA (commercial kit ELISA QUANTA LiteTM-INOVA). Test results N20 IU/mL were considered positive. APL was analyzed by ELISA (commercial kit QUANTA Lite ACA IgM III, QUANTA Lite ACA IgG III). Test results N20 MPL or N20 GPL were considered positive.

### 2.5. Specific Anti-Men-PsA, C, W_135_, Y Antibodies

Specific antibodies were analyzed by enzyme-linked immunosorbent assay (ELISA), as previously reported [5]. Briefly, 96-well plates were coated with 100 μL/well of A, C, W_135_, Y Men-Ps, kindly donated by Novartis Vaccines (now GSK), Siena, Italy, at a concentration of 10 μg/mL in phosphate-buffered saline (PBS) pH 8.2 and incubated at +4 °C overnight. After three washes with PBS supplemented with 0.005% Tween 20 (TPBS), a blocking step of 100 μL/well of TPBS with bovine serum albumin (BSA) 3% was performed for 1 h at 37 °C. After three more washes, 100 μL/well of the samples and control serum [kindly donated by Novartis Vaccines (now GSK), Siena, Italy] diluted in TPBS 1:200 was incubated for 2 h at 37 °C. After three further washes, 100 μL/well of the alkaline phosphatase-conjugated recognition antihuman immunoglobulin (Ig)G anti-serum diluted 1:1000 in TPBS were added for 1 h at 37 °C. Finally, 100 μL/well of substrate (paranitrophenil phosphate/5 mL carbonate–bicarbonate) were added and left at room temperature in the dark. The reaction was blocked by NaOH 3 M and the absorbance read at 405 nm. The concentration of Men-Ps-specific IgG in test sera was obtained by plotting the control serum dilutions (from 1:50 to 1:6400) and the corresponding IgG concentration values (μg/mL). As the titer of serum bactericidal antibodies is considered the gold standard for meningococcal seroprotection and IgG ELISA values are considered a surrogate of protection [8], we defined the IgG concentration ≥ 2 μg/mL as a ‘putative’ seroprotective level [9,10,11,12,13,14]. The seroconversion, defined as a twofold increase in IgG concentrations from T0 to T2, was considered a measure to identify subject responders to vaccination or booster doses [9].

### 2.6. Specific Anti-Tetanus/Diphtheria Antibodies

As previously reported [6], serum antibodies were examined by enzyme-linked immunosorbent assay (ELISA) commercial kits: NovaLisaTM *Clostridium tetani* toxin IgG-ELISA and *Corynebacterium diphtheriae* toxin 5S—IgG plus—ELISA (NovaTech Immunodiagnostica GmbH, Dietzenbach, Germany). Antibody concentrations 0.1 IU/mL were considered protective [15]. Subjects doubling the pre-vaccination antibody levels (ratio T2/T0 ≥ 2) were considered a responder.

### 2.7. HAV Antibody Analysis

As previously reported [7], for HAV antibody determination, a commercial ELISA kit was used: Enzygnost^®^ Anti-HAV (Siemens Healthcare Diagnostics GmbH, Marburg, Germany). The quantitative cut-off value for seropositivity was 10 mIU/mL [16].

### 2.8. Polio Antibody Analysis

#### 2.8.1. Cells and Viruses

As previously reported [7], Vero E6 cells (European Collection of Cell Cultures, Salisbury, UK) were maintained in Modified Eagle Medium (MEM) supplemented with 10% heat inactivated fetal calf serum (FCS) (Gibco, Thermo-Fisher Scientific, Waltham, MA, USA) at 37 °C in a humidified atmosphere. In our assays, the two polioviruses Sabin serotypes 1 and 3 were used (kindly provided by Dr. Giovanni Rezza from the Istituto Superiore di Sanità, Rome, Italy). The virus stocks were propagated in Vero E6 cells and were harvested at 48 h post-infection when 70–80% of cell monolayers had been killed. After freezing and thawing three times, cell lysates were clarified by low-speed centrifugation, aliquoted, and stored at −70 °C. The poliovirus Sabin type 1 virus stock was used for all the experiments at titered 107.8 50% tissue culture infectious doses per milliliter (TCID50/mL). The poliovirus Sabin type 3 virus stock was used for all the experiments at titered 106.5 TCID50/mL.

#### 2.8.2. Microneutralization Assay

The presence of poliovirus-neutralizing antibodies was assessed according to WHO guidelines [17,18]. Briefly, all serum samples, stored at −20 °C until use, were heat inactivated (56 °C for 30 min) and serially twofold diluted from 1:4 to 1:4096 in MEM containing 2% heat-inactivated FCS (final volume: 50 L) in 96-well plates. Each well dilution was challenged with 50 L of type 1 or type 3 poliovirus Sabin strain (100 TCID50 per well, 1 plate for each serotype of poliovirus) and incubated at 37 °C for 2 h in a CO_2_ incubator. After this first incubation period, each serum/viruses mix was transferred onto 24-h-old Vero E6 cells in 96-well plates and incubated in a humidified CO_2_ incubator at 36 °C for 5 days. An in-house reference serum was included in each test run to control the reproducibility of results, a 1:4 dilution of each serum without virus was included for each serum tested to check serum toxicity, and challenging viruses were back titrated in each test run to control the TCID50 challenge dose. The neutralizing antibody titer of the serum against each type of poliovirus was determined as the endpoint dilution of the serum that inhibited the poliovirus infection, observed by the cytopathic effect of inoculated cells. The neutralizing antibody assay is the method of choice to determine the immune status against poliovirus. Although the precise antibody titer that is necessary for protection is unknown, it is accepted that 1:4–1:8 of type-specific neutralization of an infection in a cell culture is putatively protective [16]; however, in the current study, the more stringent cut-off of 1:8 was chosen [19]. Responders were defined as the subjects at least able to quadruplicate baseline antibody titer at T2 [20].

### 2.9. Statistical Analysis

Categorical variables were analyzed by Yates corrected, two tails, χ^2^ test, whereas demographic data (mean age) by Student’s *t*-test [4]. Values of *p* ≤ 0.05 were considered significant. Statistical analysis was performed using the program package GraphPad Prism version 5.0 (GraphPad Software Inc., San Diego, CA, USA).

Univariate and multivariate analyses were used to evaluate factors associated with outcomes and to adjust for the effect of confounders. Multiple regression analysis was used to evaluate the effects of demographic and immunization variables on the T3/T2 ratio (expressed as the natural logarithm) of lymphocytes, total serum proteins, IgM, and IgG. The R software (The R Foundation for Statistical Computing) was used.

The persistence of vaccine-induced antibodies above the putative threshold for protection has been calculated according to the formula published by Hammarlund et al.: log(*anti-vaccine antigen antibody concentration*) = *α* + *β* × years + *ε*, where *α* represented mean log concentration at the time of vaccination; *β* represented decay rate and *ε* represented error term [21]. The decay rates (*β*) was calculated by their regression model analysis with the following equation: *β* = [log(*anti-vaccine antigen antibody concentration*) − *α* − *ε*]/years and the half-life was calculated by their formula as log(0.5)/β [21]. The durability has been estimated by the intersection of the line calculated with the equation reported above with the line indicating the threshold for protection. The vaccine-induced antibody persistence above the threshold for putative protection (2 µg/mL for meningococcal polysaccharides, 0.1 IU/mL for tetanus and diphtheria, 10 mIU/mL for HAV, a 1:8 titer for polio type 1 and 3, thus using the reciprocal 8 as the cut-off) has been calculated based on the geometric mean concentrations (GMCs) at T3 and the half-lives. In particular, the relationship between half-life and durability has been calculated by the equation *d* = *nh*, where *d* is durability in years, *h* stands for half-life and *n* is the folds half-life should be multiplied to reach the cut-off. The factor *n* may be identified by the formula *log*_2_(*x*), where *x* is the ratio *GMC/GMT at T3/cut-off for protection*. Moreover, for further control, in meningococcal polysaccharide antigens, the vaccine-induced persistence of antibody level above the putative threshold for protection was also calculated (considering a linear mean decay) using the point of intersection of the line passing through the GMCs at T2 and T3 with the threshold line for protection. The equation of the line passing through the GMCs at T2 and T3 was calculated using the formula to calculate a line having two points: GMC_T2_ (x_1_; y_1_) and GMC_T3_(x_2_; y_2_): y − y_1_ = [(y_2_ − y_1_)/(x_2_ − x_1_)] × (x − x_1_). Statistical analyses were performed by the R software (The R Foundation for Statistical Computing).

## 3. Results

Out of the military cohort originally enrolled between September 2012 and June 2014, the subjects here investigated were 112, 50 pertained to group 1, originally composed of military students, 38 males and 12 females, and 62 to the group 2, periodically engaged in a 9-month-long mission abroad (Lebanon), 60 males and 2 females (Table 1). The mean age ± standard deviation (SD) of the subjects of group 1 was 25.88 ± 2.13, with a significant difference between males (26.21 ± 2.28) and females (24.83 ± 1.11, *p* < 0.05), whereas the mean age ± SD of the subjects of the group 2 was significantly higher compared to the group 1 (34.67 ± 4.55, *p* < 0.0000001). No difference could be calculated for males and females of group 2, the latter being only 2.

The military of group 1 was only vaccinated at the time of enrolment (2012–2013), whereas 39/62 (63%) group 2 subjects had received additional vaccine boosters in the period between T2 (9-month post-vaccination) and T3 (5-year post-vaccination). In particular, the 39 group 2 subjects received a variable number of boosters, which could range from 1 to 4, so that cumulatively 14 tetanus/diphtheria, three meningococcal polysaccharides and eight meningococcal conjugated, 14 HAV, three HBV and 17 polio boosters, as well as nine oral live typhoid vaccines, have been administered in the period between T2 and T3 to these participants.

The results of non-specific parameters, such as lymphocytes, serum proteins and immunoglobulin levels at T3 compared with T2 are reported in Table 2 (upper part), whereas the relative multivariate analyses are reported in the lower part of the same Table for the subjects of group 1 and 2, respectively. Lymphocytes are significantly increased at T3 compared with T2 in group 1 in the univariate, however, this data is no more present in the multivariate analysis. Conversely, the multivariate analysis confirms the significant increase of IgM, dependent on the number of vaccines received between T2 and T3, in group 2.

No monoclonal peak in the electrophoretic pattern, nor autoimmune and/or lymphoproliferative disorders and neither other clinical adverse event have been noticed in this 5-year period. No case of hepatitis A or meningococcal meningitis has been notified to the military surveillance system in these subjects.

Autoantibodies have been investigated in all subjects and found positive in eight; six were positive for ANA, at a titer of 1:80, excepting one who had 1:160, and 2 for low-level RF (21.3 and 33.9 IU/mL, normal values ≤ 20 IU/mL). Three subjects were already positive for ANA at T0 and T2 at the same titer, including the subject with the highest titer of 1:160, whereas for the other three subjects ANA positivity was new onset, as well as for RF. The difference between T2 and T3 was not significant (Table 3).

Twenty subjects were analyzed for anti-tetanus and anti-diphtheria antibodies; of these, 7/7 (100%) of the group 1 and 12/13 (92%) of the group 2 for tetanus and 4/7 (57%) and 11/13 (85%) for diphtheria, respectively, had antibodies higher than the putative threshold for protection of 0.1 IU/mL. Out of the 60 subjects (27 of group 1 and 33 of group 2) analyzed for anti-meningococcal polysaccharide (menPs) antibodies, 27/27 (100%) and 32/33 (92%) had anti-menPsA, 18/27 (67%) and 23/33 (70%) had anti-menPsC, 22/27 (81%) and 20/33 (61%) had anti-menPsW_135_, 25/27 (93%) and 25/33 (76%) had anti-menPsY antibodies ≥ 2 µg/mL, the putative threshold for protection. Out of the 32 subjects (25 of group 1 and 7 of group 2) analyzed for anti-polio types 1 and 3 antibodies, 100% in both groups had antibodies to both polio types above the putative threshold for protection (a titer ≥1:8). Finally, 24/25 group 1 subjects had anti-HAV antibody levels above the putative threshold for protection of 10 mIU/mL. The half-life, as well as the durability above the putative threshold for protection of vaccine-induced antibodies, have been investigated in these subjects.

The group 2 subjects were chosen among those who did not have received an additional booster of the specific vaccine between T2 and T3. The GMCs at three time points (T0 = pre-vaccination, T2 = 9-month post-vaccination, T3 = 5-year post-vaccination) of the response to the tetravalent menPs vaccine, tetanus/diphtheria toxoids, HAV, and polio 1 and 3, together with the T2/T0 and the T3/T2 ratios as well as the antibody persistence above the putative protective levels (assuming a linear decay) are reported in Table 4. Despite for menPs vaccine, the military of the two groups are presented separately, this is not the case for tetanus/diphtheria and polio, for which they are considered only one group, for the small dimensions of the sample, which do not allow to appreciate any significant difference between the two groups. The anti-menPsA in the group 2 and the anti-HAV antibody persistence could not be calculated, considering that the ratio T3/T2 is ≥1.

Ten group 2 subjects, who had received at least one further booster of an adjuvanted vaccine between T2 and T3, except for MMR vaccine, were compared with 10 group 1 subjects, who did not have received any further booster in the period T2–T3; the ratio T3/T2 of antibodies addressed against measles, mumps and rubella was compared in the subjects of the two groups. No significant difference was observed for all the vaccine antigens tested (Table 5).

## 4. Discussion

The current study was carried out after a mean period of 5 years since multiple vaccinations of a cohort of military personnel. No late-appearing clinical side effect of autoimmune and/or lymphoproliferative disorders was observed during this long timeframe of follow-up, confirming our conclusions after a nine-month-long post-vaccination observation of the same cohort [4]. Positive autoantibodies have been found at low titer in eight individuals (six for ANA and two for RF), three of whom already positive at T0, with the same titer observed at T3. Out of the five individuals with new positivity, only one with a very slightly positive RF (21 IU/mL, normal values ≤ 20 IU/mL) had received four immunizations, even with adjuvanted vaccines, in the period between T2 and T3, thus in this subject only it may be hypothesized that immunization could, at least in part, be responsible for RF induction. Post-vaccine RF appearance has already been experimentally [22] and clinically [23] observed; it is generally transient and not associated with symptomatic autoimmune disease. In the remaining four subjects, the appearance of the autoantibody could not be associated with vaccination. They did not receive vaccinations in the period between T2 when they were negative, and T3. Thus, the positivity observed after 5 years should be ascribed to causes different from immunization, also considering that it is well known that the increased frequency of autoantibodies correlates with age [24], and it is not associated with autoimmune diseases. It should be emphasized that the possible dependence of autoimmunity from vaccinations is rarely documented and difficult to prove. A dependence has occasionally been suggested based on epidemiological observations, and frequently it was not confirmed by other studies. This is the case for the association HBV vaccine and multiple sclerosis [25,26,27] or type 1 diabetes and *Haemophilus influenzae* type b vaccine [28,29]. The associations between the vaccine against the human papillomavirus and autoimmune diseases, as well as the “autoimmune/autoinflammatory syndrome induced by adjuvants” (ASIA) have not been confirmed by meta-analytic studies [30,31]. Moreover, even for the association between autoimmune narcolepsy and pandemic influenza vaccine, although based on the demonstration of molecular mimicry [32], the alternative hypothesis of a direct viral cytopathic damage has been proposed [33,34]. Finally, the association between Guillain-Barré syndrome and the influenza vaccine [35], based on epidemiological observations [36], has not been systematically confirmed by subsequent studies [37]. Molecular mimicry represents the molecular basis for the association of infection/vaccination with autoimmunity. Molecular mimicry frequently occurs in nature, but control, homeostatic mechanisms prevent the development of clinical autoimmune diseases [38].

Lymphocytes, total serum proteins, and immunoglobulins, except for IgM, were not found elevated in group 2, in line with the absence of late side effects of vaccines. This observation is in line with previous reports by others [38] and confirms the results preliminarily described in this military cohort [4]. The significant increase of serum IgM, in association with the number of vaccine boosters received between T2 and T3, seems dependent on vaccine stimulation, as already observed [39], even though the increase is confined within physiological limits, likely transient, isolated, and apparently not associated to any pathology. In fact, no serum monoclonal gammopathy has been observed 5 years following the first study, despite the relatively high prevalence of the monoclonal gammopathy of undetermined significance (MGUS), which is 3% at 50 years of age, but with nearly one third estimated to have started 10–20 years before, as well as the well-known direct correlation between increased frequency of monoclonal gammopathy and increasing age [40]. Moreover, the total cohort here studied is not so small to prevent that at least one case be identified, especially because the stimuli able to induce MGUS are not well known, but an abnormal response to antigenic stimulation is nearly always present in the natural history of these patients [41].

The opportunity of studying this cohort of military personnel 5 years following multiple immunizations has also allowed calculating the half-life of the post-vaccine induced antibodies, with the consequent estimation of the antibody persistence above the putative threshold for protection. According to Amanna et al., who have carefully and deeply studied the issue of the behavior and durability of the antibody response to infections/vaccines, general kinetics of the antibody response is represented by a quick antibody decrease in the first 2–3 years since infection/vaccination, followed by a marked slowed down decrease thereafter in a slow descending or linear plateau [42]. The current study, by adding a new distant point to the monitoring of post-vaccine antibody levels, has allowed to better calculate the durability of vaccine-induced antibodies. In fact, this has been analyzed not only with the method of Hammarlund et al. [21], which has already been used in the published studies of this cohort [5,6,7] but it was also estimated (considering a linear decay) by the study of the slope of the line passing through the GMCs at T2 and T3 and the point of the intersection with the threshold line for protection. The analysis with the method of Hammarlund et al. has been calculated in two ways, as described in materials and methods. The different methods provided similar results showing an increased estimation of durability for menPs antibodies compared with our previous study [5], and a reduced estimation for tetanus, diphtheria [6], and polio 1 and 3 [7]. However, the persistence of anti-menPsA antibodies in the subjects of group 2 and of anti-HAV in those immunized with the HAV vaccine could not be calculated, because the respective curves were slightly ascending.

In our previous study, the calculated half-lives of post-vaccine anti-menPsA, C, W_135,_ and Y antibodies, were 1, 1.12, 0.91 and 1.24 years, respectively, and able to assure an antibody persistence above the putative threshold for protection ranging from 2.5 to 4.5 years [5]. In the current study, half-lives were recalculated with antibody titers measured after 5 years and found even higher, ranging from less than 2 to 8 years for different menPs and the calculated durabilities are comparable between the two methods, the first one investigating the intersection of the line of decay with the line of cut-off and the other calculating the durability from the knowledge of antibody half-life. These data are in line with the rare, published results [43,44], where the antibody durability has been either directly observed [43] or calculated with the linear regression [44]. Such a long persistence of the antibody response to polysaccharide antigens is quite surprising, considering that they are T-independent antigens unable to receive T-cell help to amplify the immune response and unable to induce memory B cells [45]. Consequently, repeated immunizations should represent brand new vaccinations and not boosters, thus carrying the risk of always recruiting new clones, which may first be expanded then induced to die for apoptosis, eventually creating the theoretical conditions for a repertoire impoverishment [46]. Thus, it seems very important to have confirmed an antibody persistence longer than the time of 3–5 years generally recommended for a new vaccine administration and calculated such persistence like the few studies on this topic [43,44]. Avoiding unnecessary boosters may contribute to maintaining the vaccine-induced antibody protection and reduce the vaccination schedules costs. The impossibility to calculate the duration of the anti-menPsA antibodies in group 2 may be a consequence of the cross-reactivity of menPsA with the polysaccharide antigens of *Escherichia coli* [47] and of *Bacillus pumilus* [48], which not-specifically may expand the immune response independently of the encounters with *Neisseria meningitidis* A, thus shaping the curve of antibody decrease linear or slightly ascending. Although the ELISA test is considered a less faithful expression of protection than the serum bactericidal titer, in adults the two assays have shown a correlation for menPsC and menPsW_135,_ but not for menPsA and menPsY [44]. Conversely, the results for the persistence of anti-tetanus toxoid and anti-diphtheria antibodies (5–6 years and 7–11, respectively) are unexpected in the light of our previous study, in which the calculation estimated persistence of 65 and 20 years for tetanus and diphtheria, respectively. The two methods based on the direct analysis of durability provided more similar and probably reliable, data compared with the method based on the identification of antibody half-life to calculate durability, in which the values were markedly low. It is not easy to provide an explanation for the discrepancy between the previous and the current study and further studies on larger populations may provide definitive data. The estimation of anti-polio antibody persistence is lower than previously observed [7], whereas for HAV the long persistence is confirmed. The long persistence of anti-HAV antibodies may be a consequence of the high immunogenicity of the HAV vaccine, but possible natural boosters cannot be excluded, since HAV is still circulating in Italy [7].

The long persistence of antibodies at putative protective levels in the absence of antigen persistence is an immunology conundrum for which different interpretations have been suggested, such as the polyclonal activation of memory B cells model, the plasma cells niche competition model, and the plasma cells imprinted lifespan model [49]. Among these, the polyclonal B cell activation theory has been proposed many times, even though the formal evidence of this mechanism is poor and only described in single case reports [50]. We had already the opportunity to explore this possible mechanism of memory maintaining in 56 individuals of our military cohort, not recently boosted for tetanus/diphtheria toxoids but recently boosted with other not correlated vaccines and analyzed with anti-diphtheria antibodies only, as a read-out system, considering that anti-tetanus antibodies may be influenced by possible accesses to the emergency room. The results did not support the thesis, since in only 4/56 subjects (7%) the specific antibodies were doubled compared to T0 levels [6]. In the current study we wanted to deepen this topic, by comparing the response to measles, mumps, and rubella antigens in 10 subjects who had, and 10 subjects who had not, received vaccine boosters other than MMR between T2 and T3. The comparative analysis did not show any significant difference between the two groups. Despite menPs vaccines are considered unable to induce memory B-cells, the immune response to this vaccine we previously observed allowed us to hypothesize that memory B-cells could actually have been induced [5], thus we decided to monitor the response to menPsA, C, W_135_ and Y in 11 subjects who had received vaccine boosters other than tetravalent menPs vaccine between T2 and T3, compared with 11 control subjects. Even in this case, no evidence of polyclonal B cell activation has been observed. Even if the number of analyzed cases is limited, it seems that polyclonal activation of memory B-cells is at least not frequently involved in maintaining post-vaccine antibody levels in normal adult people who had received multiple immunizations with these traditional vaccines. This result agrees with what has been observed by Amanna et al. [51] on four adults, who received a smallpox vaccine booster and have been monitored for one year post-vaccination for pre-existing antibodies against nine antigens (vaccinia, tetanus, diphtheria, and pertussis toxoids, measles, mumps, rubella, varicella, and Epstein Barr virus). Vaccinia antibodies did increase 8-80-fold, while antibodies against the other eight antigens did not.

The strength of this study is that we have confirmed the lack of late clinical and laboratory side effects in a military cohort followed up for 5 years from vaccinations and calculated the persistence of vaccine-induced antibodies at putative protective levels, confirming a theoretical quite long, sometimes life-long, protection towards the different vaccine antigens, including the menPs vaccine antigens, for which values higher than previously identified have been calculated [5]. The documented safety at 5-year post-vaccination is a relevant message, considering the scarcity of systematic studies on such topic, and particularly in the current pandemic period, in which the fear of late side-effects by anti-coronavirus disease (COVID)-19 vaccines in a significant percentage of the global general population risks hindering or slowing down public health interventions. The main weakness is related to the absence of blood samples to be tested at different times, to make the calculation of the post-vaccine antibody durability more precise.

In conclusion, it is confirmed that multiple vaccinations in the military are not accompanied by side effects characterized by autoimmunity/lymphoproliferation, irrespective of the type of operativity, even at 5 years since multiple immunizations. The calculated vaccine-induced antibody durability is quite long, also in response to vaccines composed of T-independent antigens, unable to induce memory cells, such as menPs vaccine. Such a long antibody persistence does not seem to be a consequence of a polyclonal B cell activation.

## Figures and Tables

**Table 1 biomedicines-10-00006-t001:** Demographic data of the military of group 1 and group 2.

Military Subjects	N (%)	M/F	Mean Age ± SD
Group 1	50 (45)	38/12 °	25.88 ± 2.13 *
Group 2	62 (55)	60/2	34.67 ± 4.55
Total	112 (100)	98/14	30.34 ± 5.46

° *p* = 0.002550 vs. group 2; * *p* < 0.0000001 vs. group 2.

**Table 2 biomedicines-10-00006-t002:** Univariate analysis of mean values ± SD of peripheral blood lymphocytes, serum protein and immunoglobulin measures at two time points in group 1 and 2 subjects, and Multivariate analysis of demographic and immunization variables on lymphocytes, total proteins, IgM, and IgG levels in both military groups.

**Univariate Analysis**	**Group 1**								**Group 2**							
	**IgG** **(mg/dL)**	**IgM** **(mg/dL)**		**IgA** **(mg/dL)**		**Lymphocytes (cells/µL)**	**Total** **Proteins** **(g/dL)**		**IgG** **(mg/dL)**		**IgM** **(mg/dL)**		**IgA** **(mg/dL)**	**Lymphocytes (cells/µL)**	**Total Proteins** **(g/dL)**	
	Mean values ± SD								Mean values ± SD							
**T2**	1142 ± 187	115 ± 49		194 ± 64		1969 ± 452	8.03 ± 1.05		1092 ± 329		95 ± 32		233 ± 90	2247 ± 647	6.29 ± 0.35	
**T3**	1095 ± 287	115 ± 40		199 ± 76		2255 ± 529	8.08 ± 0.77		1051 ± 288		128 ± 60		242 ± 87	2398 ± 591	7.34 ± 0.91	
** *p* **	ns	ns		ns		<0.01	ns		ns		<0.02		ns	ns	<0.000001	
**Multivariate Analysis**	**Group 1**								**Group 2**							
	**IgG** **(°)**		**IgM** **(°)**		**Lymphocytes (°)**		**Proteins** **(°)**		**IgG** **(°)**		**IgM** **(°)**		**Lymphocytes (°)**		**Proteins** **(°)**	
	**b-Coefficient**	** *p* **	**b-Coefficient**	** *p* **	**b-Coefficient**	** *p* **	**b-Coefficient**	** *p* **	**b-Coefficient**	** *p* **	**b-Coefficient**	** *p* **	**b-Coefficient**	** *p* **	**b-Coefficient**	** *p* **
**Diftetal**	0.2442	ns	−0.1851	ns	0.4117	ns	0.0296	ns	−0.0520	ns	−0.2864	ns	0.01530	ns	−0.04774	ns
**Engerix**	/	/	/	/	/	/	/	/	0.0255	ns	0.1231	ns	−0.004031	ns	/	/
**Epaxal**	0.2025	ns	0.0004	ns	0.2857	ns	0.0260	ns	0.0645	ns	−0.04155	ns	0.04929	ns	−0.0111	ns
**Age**	0.0031	ns	0.0091	ns	−0.0028	ns	−0.0006	ns	0.0043	ns	0.008891	ns	0.000916	ns	−0.0067	ns
**Imovax polio**	0.4479	0.009	−0.3151	ns	0.4282	ns	0.02207	ns	0.08944	ns	0.05133	ns	0.008030	ns	0.03883	ns
**Sex (M/F)**	0.0713	ns	−0.02695	ns	−0.0187	ns	−0.02494	ns	−0.03510	ns	0.008077	ns	0.07032	ns	/	/
**Mencevax**	0.3139	ns	−0.2176	ns	0.09243	ns	0.02605	ns	−0.06887	ns	−0.04949	ns	−0.06048	ns	0.0329	ns
**N vax**	−0.3036	ns	0.1956	ns	−0.3932	ns	−0.04185	ns	−0.04361	ns	0.02901	ns	−0.02765	ns	−0.1037	ns
**N vax after T2**	/	/	/	/	/	/	/	/	−0.01710	ns	0.1264	0.012	−0.02657	ns	−0.0089	ns
**Priorix**	−0.0650	ns	0.0954	ns	0.2247	ns	0.01782	ns	/	/	/	/	/	/	/	/
**Varivax**	0.5520	0.009	−0.0210	ns	0.4997	ns	/	/	0.01169	ns	−0.2008	ns	−0.04840	ns	/	/
**Vivotif**	/	/	/	/	/	/	/	/	0.04287	ns	0.05843	ns	−0.003886	ns	−0.1020	ns

(°) Natural logarithm of the ratio (T3/T2).

**Table 3 biomedicines-10-00006-t003:** Autoantibodies in 112 military subjects.

Autoantibodies	N (%) of Positive at T2	N (%) of Positive at T3	*p*
ANA	3 (2.67)	6 (5.36)	NS
RF	0	2 (1.78)	NS
Total	3 (2.67)	8 (7.14)	NS

ANA = anti-nuclear antibodies; RF = rheumatoid factor.

**Table 4 biomedicines-10-00006-t004:** Geometric mean concentrations of anti-meningococcal polysaccharides (menPs) A, C, W_135_, Y, anti-tetanus, and diphtheria toxoids, anti-hepatitis A virus (HAV), and geometric mean titers of anti-polio 1 and 3 in the military of both groups pre- (T0), 9-month (T2) and 5-year (T3) post-vaccination, as well as the T2/T0 and T3/T2 ratios, are reported. Half-life of vaccine-induced antibodies, with their 95% confidence interval (CI), as well as antibody durability above the putative threshold for protection in years are also indicated.

Antigen	Group	T0	T2	T3	T2/T0	T3/T2	Half-Life (Years)	95% CI	Durability *		
									A	B	C
menPsA	1	0.65	32.94	18.02	50.68	0.55	4.92	3.42–8.42	15.60	20.60	9.30
menPsA	2	4.93	27.44	28.60	5.56	1.04	NC	NC	NC	NC	NC
menPsC	1	0.07	11.96	6.43	170.86	0.54	4.75	3.25–8.58	8.00	13.00	8.40
menPsC	2	9.11	15.49	4.97	1.70	0.32	2.58	2.00–3.58	3.38	8.42	6.2
menPsW_135_	1	0.37	5.83	3.06	15.76	0.52	4.50	2.67–15.00	2.74	7.50	6.90
menPsW_135_	2	5.91	9.85	3.50	1.67	0.35	2.83	2.17–4.25	2.28	7.30	6.00
menPsY	1	0.38	8.81	5.88	23.18	0.67	8.00	3.58–∞	12.40	17.08	11.20
menPsY	2	3.38	13.56	2.12	4.01	0.15	1.58	1.17–2.50	0.13	5.16	5.00
Tetanus	1 + 2	1.44	4.00	0.25	2.78	0.06	1.08	1.00–1.08	1.43	6.42	5.00
Diphtheria	1 + 2	0.13	0.53	0.24	4.00	0.45	4.25	3.33–5.92	3.15	11.58	7.00
Polio 1	1 + 2	48.00	359.00	172.00	7.50	0.48	2.50	2.17–2.92	11.00	17.33	8.70
Polio 3	1 + 2	33.00	546.00	330.00	16.50	0.60	3.08	2.58–3.75	16.53	23.25	11.30
HAV	1	0.26	44.18	48.76	170.00	1.10	NC	NC	NC	NC	NC

* The durability is expressed in years: in the column A it has been calculated by the equation *d* = *nh*, where *d* is durability, *h* stands for half-life and *n* is the folds half-life should be multiplied for to reach the cut-off; in the column B by estimating the linear regression of the equation log(antibody titer) = *α* + *β* × years + *ε* and calculating the intersection with the line of the threshold for protection; in the column C the durability has been estimated by the intersection of the line joining the geometric mean concentrations/titers at T2 (9-month) and T3 (5-year) with the line of the threshold for protection. NC = non-calculable.

**Table 5 biomedicines-10-00006-t005:** Analysis of 10 subjects of group 2 who had received boosters of adjuvanted vaccines between T2 and T3 compared with 10 subjects of group 1 who did not receive boosters.

Antigen	Measles T3/T2	Measles T3/T2	Mumps T3/T2	Mumps T3/T2	Rubella T3/T2	Rubella T3/T2
Military group	1	2	1	2	1	2
Subject 1	1.23	2.56	1.19	1.34	0.67	2.42
Subject 2	0.49	1.87	0.75	1.76	0.89	1.49
Subject 3	0.78	2.22	0.97	1.95	0.89	4.50
Subject 4	0.47	0.93	0.52	1.16	0.53	0.87
Subject 5	0.84	0.73	0.81	0.85	1.06	0.86
Subject 6	0.77	1.54	0.59	1.08	0.65	1.00
Subject 7	0.96	1.05	0.64	1.20	0.57	0.92
Subject 8	1.00	0.38	1.85	1.32	0.56	0.33
Subject 9	0.94	0.04	1.28	1.00	0.40	0.33
Subject 10	1.43	0.52	0.55	0.19	0.59	0.15

Χ^2^ between group 2 and group 1 by considering positive those with a T3/T2 ≥ 2 = NS for all the vaccine antigens.

## Data Availability

Data supporting reported results may be found at Sapienza, Università di Roma, Dipartimento di Medicina Clinica e Molecolare, Dipartimento di Medicine Molecolare, Dipartimento di Medicina Sperimentale, Dipartimento di Sanità Pubblica e Malattie Infettive, at Istituto Superiore di Sanità, Dipartimento di Malattie Infettive, Parassitarie e Immuno-mediate and at Laboratorio di Virologia, IRCCS, Istituto Nazionale Malattie Infettive “Lazzaro Spallanzani”.

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
