# Peer review of "Safety of Multiple Vaccinations and Durability of Vaccine-Induced Antibodies in an Italian Military Cohort 5 Years after Immunization"

_biomedicines, 2021, doi:10.3390/biomedicines10010006_

Round 1

Reviewer 1 Report

The manuscript is devoted to the study of safety of several vaccines and durability of specific antibodies induced by them. The importance of the study is undeniable. The amount of data analyzed is admirable.

Nevertheless, there are some points to be specified:

  1. The research is quite complex with plenty of parametres analized. Perhaps, it would be better to split it in two papers.
  2. Materials and Methods. Statistic analysis. The formula of the vaccine-indused antibody persistence calculation should be added.
  3. The relationship of mean antibody durability and antibody half-life can be clarified.
  4. The increasing of the calculated half-life of antibodies to meningococcal polysaccharides (espessially for anti-PsA antibodies) is surprising. Are there any data from outside groups on the antibodies persistence after meningococcal vaccination to compare?
  5. The efficacy of some vaccines is conditioned by not only antibodies but cellular immune responses. If there was any information about T cell reactions in participant of this research it would be interesting to see it, probably within a separate article.

Reviewer 2 Report

The authors used a prospective study to address the effect of vaccination on durability of antibodies. Because vaccination is one of the essential public health countermeasures against infectious diseases, it would be important to identify its durability. Authors suggest that there is a long half-life for anti-HAV, polio 1 and 3, tetanus and diphtheria. However, this article has not been fully answered some of questions due to the insufficient description.

First, authors used a lot of tables in their article, but these tables should be combined for easy understand by readers. For example, I recommend that table 2-4 as well as table 6-8 should be combined. Moreover, I could NOT understand why figure 1 and figure 2 were needed. If these figures are NOT needed, authors should delete these figures.

Second, there is no description of statistical models for multivariable analysis of table 2 in materials and methods section. It is impossible to understand the results without description of statistical analysis. Authors should add the description of the statistical analysis of this multivariable analysis in materials and methods section.

Third, the description in 2.9. Statistical analysis is poor. For example, there is no explanation of α and ε, and the description “(β=[log(antibody concentration –α-ε]/years [21]” do NOT have “)”. Authors should be carefully about the description of statistical analysis.

Finally, the paragraphs in discussion section are too long to read. I suggest that the discussion should be organized to shorten the paragraphs.

Reviewer 3 Report

The paper of Claudia Ferlito and co-authors analyzes the safety of multiple vaccinations and durability of vaccine-induced antibodies in an Italian military cohort 5 years after immunization. 

The data itself and results of the analysis mostly repeat already published finding (references 4-7 of the reviewing manuscript). I do not recommend publishing the manuscript due to lack of novelty replaced with irrelevant data: 

1) It's hard to believe that the lack of evidence for post-vaccine onset of autoimmune disorders during a nine-month follow-up won't be confirmed 5 years later. 

2) The results of non-specific parameters reported in tables 2-4 are irrelevant and just inflate the size of the manuscripts; 

3) Authors even did not try to explain the difference between groups 1 and 2 reflected in Fig. 1 A-b and C-D, which is weird. 

Round 2

Reviewer 2 Report

The authors revised their manuscript in response to my comments, and I have no further comments.